# Bench, Bar, and Ring Dips: Do Kinematics and Muscle Activity Differ?

**DOI:** 10.3390/ijerph192013211

**Published:** 2022-10-14

**Authors:** Alec McKenzie, Zachary Crowley-McHattan, Rudi Meir, John Whitting, Wynand Volschenk

**Affiliations:** Department of Sport and Exercise Science, Faculty of Health, Southern Cross University, Lismore, NSW 2480, Australia

**Keywords:** the dip, exercise technique, exercise prescription, electromyography, kinematics

## Abstract

The purpose of this study was to profile and compare the kinematics, using 3D motion capture, and muscle activation patterns, using surface electromyography (sEMG), of three common dip variations; the bench, bar, and ring dips. Thirteen experienced males performed four repetitions of each dip variation. For each participant, repetitions 2–4 were time-normalized and then averaged to produce a mean value for all kinematic and sEMG variables. The mean maximal joint angles and mean peak sEMG amplitudes were compared between each variation using a one-way ANOVA with repeated measures. Several significant differences (*p* < 0.05) between dip variations were observed in both kinematic and sEMG data. The bench dip predominantly targets the triceps brachii but requires greater shoulder extension range. The mean peak triceps brachii activation was 0.83 ± 0.34 mV on the bench, 1.04 ± 0.27 mV on the bar, and 1.05 ± 0.40 mV on the ring. The bar dip is an appropriate progression from the bench dip due to the higher peak muscle activations. The ring dip had similar peak activations to the bar dip, with three muscles increasing their activation intensities further. These findings have implications for practitioners prescribing the dip, particularly to exercisers with a history of shoulder pain and injury.

## 1. Introduction

The dip exercise is commonly prescribed across a wide range of training contexts. It involves an exerciser supporting their bodyweight through the upper limbs whilst grasping equipment such as a weightlifting bench, parallel bars, or gymnastics rings. The exerciser then lowers the body by flexing the elbows and extending the shoulders before returning to the starting position by extending the elbows and flexing the shoulders [1,2].

As an exercise, it is generally thought to be effective for strengthening the shoulder complex and upper limb muscles, particularly the triceps brachii (TB) and pectoralis major (PM) [1,2]. The dip’s popularity stems from its adaptability to suit a wide range of sport-specific movement patterns, i.e., push movements, and its versatility in training. The three more common dip variations seen in practice may be the bench dip, bar dip, and ring dip (Figure 1).

The dip is often prescribed in a wide range of exercise contexts, from increasing sports performance in experienced athletes [1,2,3], to athletes rehabilitating from shoulder instability [4] or other upper extremity injuries [5,6]. For beginners, the bench dip may be an appropriate starting point to introduce the dip because the exerciser can moderate the resistance by placing their feet on the ground. More experienced exercisers often choose to overload the shoulder flexors and elbow extensors by using the body-weighted bar dips. The bar dip has previously been used to assess and develop both muscular endurance [3] and maximal upper-body push strength [1,2]. More recently, ring dips have been popularised as another dip variation primarily aiming to increase the stabilization requirements and thereby increase muscular demands of the upper limb and trunk while performing the movement [7]. Further to these examples, the dip (no specified variation) has been used in physician’s protocols to rehabilitate TB [8] and PM [9] ruptures and improve swimming performance [10]. Yet, the prescription of the dip in these various performance and rehabilitation programs is based on the practitioner’s experience rather than empirical evidence. While other upper-body push exercises, that may have similar neuromechanical profiles, have previously been investigated, including the push-up [11,12,13], bench press [14], and shoulder press [15,16,17], such research has not been conducted on dip variations, and so the neuromechanical profile of these exercises remains unknown. 

In addition to the lack of evidence justifying the dips use in exercise programs, the dip has also been criticised as a potentially high-risk exercise for injury to the shoulder [2,7,18]. A potential mechanism of injury has been suggested by McKenzie, Crowley-McHattan [18] relating to injury of anterior shoulder and PM. However, supporting evidence is limited, with only two case reports of PM ruptures caused by the bar dip [19,20], anecdotal examples of PM injury caused by the ring dip [21,22], and unsubstantiated claims of the ring dip being a high-risk exercise for shoulder injury [7]. These injury concerns may be somewhat addressed with a deeper understanding of the movement profile of the dip.

Curiously, despite its widespread popularity, and the anecdotal high injury risk, the dip has received little attention in the literature. Therefore, this investigation aimed to profile and compare the kinematics and muscle activation patterns of three common dip variations, i.e., the bench, bar, and ring dips. We hypothesized that the three dip variations would use significantly different movement kinematics and muscle activity profiles for the upper limb and trunk. The systematic investigation of the dip’s movement characteristics and muscle activation patterns will help exercise professionals understand the appropriate prescription of each dip variation in practice and the aid the understanding of potential injury risks. 

## 2. Materials and Methods

### 2.1. Participants

An a priori power calculation was conducted, and it was found that a minimum sample of five participants was needed based on a power of 0.8 and alpha level of 0.05. Therefore, a total of 13 males were recruited from the university and local gyms to participate in this cross-sectional observational research. All participants offered informed consent and volunteered for this study. The mean age was 29.92 ± 6.66 years, with a mean resistance training age of 4.16 ± 5.56 years (height = 171.68 ± 29.32 cm, weight = 88.96 ± 28.78 Kg). All participants were experienced with dip variations, regularly incorporating bodyweight dips in their weekly structured strength and conditioning programming. Participants were excluded if they had experienced any chest or upper extremity injuries in the prior 12 months. Two participants did not have regular experience with the ring dip and were excluded from this variation. The study was approved by the institution’s Human Research Ethics Committee (ECN-19-223).

### 2.2. Procedures

The University of Western Australia’s full-body model [23] was used to collect 3D kinematics of the entire body during each dip variation using 14-3D cameras (Vicon, Oxford, UK). Kinematic data were sampled at 200 Hz to determine the peak angles of elbow flexion, anterior thoracic lean (relative to vertical), and shoulder extension, abduction, and external rotation. The vertical displacement of the pelvis was measured to differentiate repetitions.

Wireless sEMG electrodes (Delsys Avanti Wireless, Natick, MA, USA) were used to record muscle activation profiles. Nine muscles on the participant’s right arm and trunk were collected following standardized electrode placements [24]. The sEMG data were sampled at 2000 Hz and synchronized with the kinematic data using Nexus data collection software (Nexus 2, Oxford, UK). The peak activation amplitudes were recorded for nine muscles which were chosen for the suspected roles as: (i)Potential agonists of shoulder flexion and elbow extension: PM clavicular head, anterior deltoid (AD) and TB lateral head,(ii)Potential active scapular stabilizers: upper trapezius (UT), serratus anterior (SA) and lower trapezius (LT), and(iii)Potential active glenohumeral stabilizers: infraspinatus (IS), latissimus dorsi (LD) and biceps brachii (BB).

All dip variations were performed on a free-standing matrix frame (Iron Edge, Victoria, Australia) located in the center of the university biomechanics laboratory. A “Side Load Farmers Walk Handle” bar (Iron Edge, Victoria, Australia) was used at the height typical for a bench dip. A V-shaped dip bar (Iron Edge, Victoria, Australia) was mounted directly to one post of the matrix rack at a self-selected height and used for bar dips. For the ring dips, 28 mm wooden power rings (Iron Edge, Victoria, Australia) were anchored 3.5 m above the ground, with the participant self-selecting the height above ground. The dip bar and rings were typically mounted from waist-to-shoulder height. Figure 1 depicts the typical body position used by a participant in each dip variation.

### 2.3. Data Collection

All participants attended the laboratory on one occasion for 1–2 h. Participants were first familiarized with the exercise equipment and upcoming protocols. Next, retroreflective markers and sEMG electrodes were fixed to the skin, and a standardized warm-up consisting of 15 arm swings, 10 push-ups and 5 rows was completed [25]. Immediately before data collection, each participant performed a maximal shoulder extension range of motion (ROM) test. The dip bar was placed at approximately waist height for this test, with the participant’s feet grounded. The participant was directed to lower themselves as far as possible, as if they were completing a bar dip, then to hold the bottom position for three seconds before returning to the standing position. Three repetitions were performed with the maximal value used as an indicator of self-moderated maximal shoulder extension ROM. Each participant then completed four repetitions of the three dip variations in a counterbalanced order, such that at least two participants completed each of the six possible ordering sequences. Participants were given 2–4 min of recovery between variations to avoid cumulative fatigue. No coaching cues were provided at any point, allowing each participant to adopt a self-moderated technique and ROM for each dip variation.

### 2.4. Data Analysis

To identify and differentiate each repetition, the 3D mid-pelvis position was determined geometrically from adjacent markers, and its height was used to define each repetition and repetition phase. The frame before the mid-pelvis began lowering was deemed the start of the repetition. The time point coinciding with the lowest mid-pelvis position was deemed the bottom position of the dip and the end of the downwards phase. The following peak in mid-pelvis height was used to define the end of the upwards phase and the end of a repetition. 

Raw kinematic data were processed using a fourth-order Butterworth filter. Raw sEMG data for each muscle were smoothed and rectified using a root mean squared (RMS) algorithm with a moving average of 0.4 s. Repetitions 2–4 were time-normalized with 200 data points calculated each representing 0.5% of a complete repetition cycle. Next, the average of repetitions 2–4 was calculated at each of these normalized time points and then used for statistical analysis for the variables listed previously. The normalization procedure is displayed as a flow chart in Figure 2.

### 2.5. Statistical Analysis

All data were checked for normality using the Shapiro–Wilk test and found to be normally distributed. One-way repeated measures ANOVA were used to compare kinematic and sEMG variables within-participants between dip variations. Where a significant main effect was found, pairwise comparisons with Bonferroni correction were made. Significance was set at an alpha level of 0.05. Effect sizes for dip condition were calculated using partial Eta squared (η_p_^2^), with Cohen’s d being used to measure the effect size on pairwise comparisons (*d* < 0.2 small effect, *d* < 0.5 medium, *d* < 0.8 large, *d* < 1.2 very large effect [26]). All statistics were calculated using SPSS version 25 (IBM, Armonk, NY, USA).

## 3. Results

### 3.1. Repetition Characteristics

A significant effect of dip condition (dip variation) was found for vertical displacement of the mid-pelvis (*p* = 0.002, η_p_^2^ = 0.758). The bar dip (394.30 ± 73.32 mm) had a significantly greater vertical displacement of the mid-pelvis compared to the bench dip (318.18 ± 85.60 mm. *p* = 0.008, *d* = 0.93) and ring dip (339.31 ± 73.42 mm, *p* = 0.001, *d* = 0.75). However, the bench and ring dips were not significantly different (p = 0.835, d = 0.26). No main effect was found for repetition duration (*p* = 0.417, η_p_^2^ = 0.177, bench = 2.17 ± 0.40 s, bar = 2.32 ± 0.43 s, and ring = 2.31 ± 0.61 s), or for relative timing of the bottom position within the repetition (*p* = 0.218, η_p_^2^ = 0.316, bench = 58.56 ± 3.83%, bar = 57.69 ± 5.37%, and ring = 55.69 ± 3.83%).

### 3.2. Kinematics

The mean peak joint angles for each dip variation are displayed in Figure 3. A significant condition effect was found for shoulder extension (*p* < 0.001, η_p_^2^ = 0.824), anterior thoracic lean (*p* < 0.001, η_p_^2^ = 0.889) and elbow flexion (*p* = 0.025, η_p_^2^ = 0.561) but not for shoulder abduction (*p* = 0.516, η_p_^2^ = 0.137) or external rotation (*p* = 0.445, η_p_^2^ = 0.165). Shoulder extension (*p* = 0.008, *d* = 1.06) was significantly greater, with anterior thoracic lean (*p* < 0.001, *d* = 3.46) and elbow flexion (*p* = 0.026, *d* = 1.38) being significantly less during the bench dip compared to the and bar dip. Shoulder extension (*p* < 0.001, *d* = 2.31) was significantly greater and anterior thoracic lean (*p* = 0.005, *d* = 1.73) was significantly less during the bench dip compared to the ring dip. Shoulder extension (*p* = 0.001, *d* = 1.39) and elbow flexion (*p* = 0.031, *d* = 0.93) were significantly greater during the bar dip compared to the ring dip. 

During the maximal ROM trial, participants had an average maximal shoulder extension angle of 88.72% (±9.38°). When displayed as a percentage of the maximal ROM test, peak shoulder extension was 101.35% (±6.49%) during the bench dip, 88.03% (±10.20%) during the bar dip, and 68.88% (±13.88%) during the ring dip.

### 3.3. Muscle Electromyography (Activations)

There was a significant effect of condition for the peak activation amplitude of eight muscles investigated (PM, AD, TB, UT, SA, IS, LD, and BB) (*p* > 0.001–0.030, η_p_^2^ = 0.541–0.915). No significant effect of condition was found for LT (*p* = 0.135, η_p_^2^ = 0.359). The average peak activation amplitude for all muscles in each variation are illustrated in Figure 4. Six muscles (PM, AD, TB, UT, SA, and LD) exhibited significantly greater peak activation on the bar dip compared to the bench dip (*p* < 0.001–0.008, *d* = 0.66–3.37). Eight muscles (PM, AD, TB, UT, SA, IS, LD, and BB) had significantly greater peak activation intensity on the ring dip compared to the bench dip (*p* < 0.001–0.026, *d* = 0.58–3.32). Only three muscles displayed significantly different peak activations between the bar and ring dips (PM, LD and BB), all exhibiting their greatest activation during the ring dip (*p* = 0.002–0.008, *d* = 0.73–0.78). The LT was not significantly different between each variation (*p* = 0.294->0.999).

## 4. Discussion

Overall, the kinematic characteristics of the bench dip were largely different to the bar and rind dips. This was accompanied by significantly lower activity of most muscle groups investigated compared to the bar and ring dips. Therefore, these findings partially support our hypothesis that there would be significant differences between dip variations in relation to kinematic and muscle activation profiles.

### 4.1. The Bench Dip

The bench dip required relatively high muscle activity of the TB compared to other muscles and used a much greater shoulder extension ROM compared to other dip variations. The small activation intensity is likely due to the reduction in load and greater stability due to the foot-ground contact [27]. While the peak activation intensity of TB is significantly less than during the bar and ring dips, the bench does target the TB relatively well when considering the reduced resistance through the upper limb, the overall movement complexity, and the relatively low activation of other prime movers (PM and AD). The dip is often called the “triceps dip” [4,8], which appears appropriate for the bench dip but remains unclear for other variations according to this investigation’s findings. Previous research has compared the activation intensity of TB when completing eight common TB exercises and concluded that the bench dip, triangle push-up and triceps kickback exercises exhibited the equally greatest TB activation intensity [28]. Therefore, the triangle push-up and triceps kickback exercises may be good alternatives for individuals aiming to train TB, with shoulder instability issues.

The bench dip also had the largest shoulder extension ROM, which was on average 88.13° (±8.86°), representing 101.35% (±6.49%) of the maximal ROM test. This much larger ROM is likely due to the smaller amount of anterior thoracic lean (12.61° ± 6.49°) compared to both the bar and ring dip variations. This results in a larger proportion of the bench dip depth from shoulder extension rather than being distributed through anterior thoracic lean and shoulder extension. This greater ROM at the shoulder joint may increase the risk of injury to the anterior shoulder capsule through the repetitive submaximal straining of the anterior band of the inferior glenohumeral ligament. This ligament has shown to be strained in glenohumeral extension [29], and to become permanently elongated with repetitive submaximal strains [30], similar to that observed during the bench dip. The elongation of this ligament is likely to present as anterior shoulder instability [31]. This larger shoulder extension ROM may also excessively strain the PM when likely operating at a mechanical disadvantage [18,32]. However, the smaller load and low intensity of PM activation may mitigate PM injury risk. Rather, the greater concern is the potential for damage to the anterior shoulder capsule, perpetuating or producing anterior shoulder instability.

### 4.2. The Bar Dip

The bar dip is seen as a progression from the bench dip as there is greater total load and a reduction in movement stability through the loss of the foot-ground contact points. This common practice is supported by the significant increase in peak activation amplitudes of most muscles tested (PM, AD, TB, UT, SA, and LD) compared to those seen in the bench dip (Figure 4). Interestingly, despite the increased load and stability requirements, two of the suspected glenohumeral stabilizers, IS and BB, did not increase their peak activation intensity. Previous research has found that while small decreases in stability may increase muscle activations, larger increases lead to a loss of force output and reduction in muscle activation [33].

The bar dip also had the greatest mid-pelvis vertical displacement despite having significantly less peak shoulder extension than the bench dip. In the bar dip, the thorax can anteriorly lean, which aids the depth of the movement while reducing the total proportion of depth arising from shoulder extension. However, the extension used was 88.03% (±10.20%) of the maximal ROM demonstrated by the maximal ROM test, which may still be loading the strained inferior glenohumeral ligament, thereby increasing the risk of anterior instability. In addition to this, the increased resistance and the significant increase in PM activation must be considered with respect to PM injury, particularly when considering prior case reports of PM rupture occurring during the bar dip [19,20]. The body position used by the participants of this study under the test conditions of low repetitions (four per exercise), bodyweight only, minimal fatigue (low total repetitions compared to a normal training load), and a V-shaped dip bar (typically less shoulder abduction compared to parallel bars), appeared to be beneficial in terms of safely loading the passive and active structures of the shoulder. However, if any of these conditions are altered (e.g., additional load, increased fatigue or wide-grip hand position), the risk of PM injury may markedly increase. Further investigation is warranted since the bar dip may be considered the most common dip variation [1].

When considering prescribing bar dips to an exerciser, potentially compromised by inexperience, weakness or injury, it is important for practitioners to understand that a body-weighted bar dip, as conducted in this investigation, may overload passive and active shoulder structures. Regression modifications can be made to allow for a similar body position (more optimal than the bench dip), while progressively increasing total load through the upper limb. Practitioners may consider standing bar dips for highly in-experienced exercisers, which would allow them to self-moderate the load through their lower limbs. If accessible, assisted bar dips (using a dip machine) may also be beneficial for all stages of progression, as counter-resistance can be decreased as strength increases. More experienced exercisers may be better advised to use banded-bar dips. This would likely increase the stability requirements while still decreasing the load through the upper limb, particularly during the bottom position where the band would provide the most resistance. These three regressions are illustrated in Figure 5. 

### 4.3. The Ring Dip

Despite the considerable increase in the ring dip’s movement complexity, only three muscles increased their peak activation compared to the bar dip; these were the PM, LD and BB. We postulate that the increases in PM and LD mainly acted to adduct the arm, thus helping to increase shoulder joint stability, while the BB was primarily acting to stiffen the elbow joint and to maintain glenohumeral joint congruity. This indicates that the scapula and glenohumeral stabilization requirements are similar between bar and ring dips when conducted by experienced exercisers, despite the difference in perceived movement complexity and peak shoulder extension. 

The ring dip required a very similar body position to the bar dip, with only peak shoulder extension and elbow flexion decreasing during the ring dip. The peak extension angle was 61.72° (±13.51°), representing 68.88% (±12.88%) of the maximal ROM test and was the smallest of all dip variations. Intuitively there are perhaps the greatest concerns for injury while performing the ring dip. However, this variation utilized the least amount of shoulder extension, which was previously identified as the dip’s most vulnerable position [18]. As the technique was entirely self-moderated, participants were potentially avoiding the vulnerability of end-range extension whilst performing a less stable task, which would likely have reduced end-range loading in the more unstable condition. Unlike the bench and bar variations, the primary concerns of the ring dip should not be the stresses and strains experienced through the shoulder complex, but more so the risk of traumatic injuries occurring due to falling from the rings [7].The findings of this study should be considered when prescribing the dip in performance and rehabilitation contexts, particularly to those with a history of shoulder pain and injury. However, the main limitation in the practical application of this investigation’s findings is that the technique used by participants was entirely self-moderated. In practice, there are a number of specific, and differing, coaching cues used to stipulate correct completion of a dip [1,3]. These may alter the overall movement characteristics and muscle activations during the repetition; further research is warranted investigating specific coaching cues. Additionally, this investigation only analyzed single points of data within the entire repetition cycle. Future investigations should consider using continuous data analysis methods such as Statistical Parametric Mapping or Principal Component Analysis to investigate potential differences at other points throughout the complete repetition. Further, future studies may also consider assessing the effects of movement velocities and accelerations on muscle activity and joint and tissue loading. Finally, our sample was relatively small consisting only of healthy males, aged 20-40 yrs (mean =29.92 ± 6.66 yrs). It is possible that healthy joint ranges may differ in older individuals, females, or exercisers with hypo- or hyper-mobility conditions. In these exercisers, a cautious approach should be taken at end-range, until future research investigates these cohorts further.

A possible decision tree has been created in Figure 6 to assist practitioners decision to prescribed on dip variation over the other.

## 5. Conclusions

The bench dip required a relatively high TB activation intensity when considering exercise load and complexity. However, participants of this study used beyond their observed self-moderated maximal shoulder extension, which may increase the risk of shoulder injury. As the TB can be activated to similar levels in other exercises that do not require loaded end-range extension, the bench dip should not be part of regular strength training or rehabilitation protocols.

The bar dip used a more optimal body position compared to the bench dip and appeared to be an appropriate exercise progression due to the increased muscle activity overall. If load at the point of maximal extension is modifiable, the bar dip may also be an appropriate exercise for rehabilitation of the anterior shoulder. This includes regression variations (examples displayed in Figure 5) and progression variation (where external load can be suspended by the waist). These modifications should provide a progression model that can appropriately increase the load in the active and passive stabilizing structures.

The ring dip required the greatest peak activation of three muscles, PM, LD and BB, and required the smallest peak shoulder extension. If the fall risk is of concern (e.g., in novice athletes or during fatigue), exercise professionals should consider prescribing the bar dip instead; otherwise, the ring dip may be the best dip variation when targeting the PM.

The dip has been prescribed for four reasons; to train the TB, to train the PM, to increase upper body push strength, or to rehabilitate or “prehabilitate” from upper extremity injury. Figure 6 is a flow chart that illustrates which dip variation is recommended for use in each of these given purposes.

## Figures and Tables

**Figure 1 ijerph-19-13211-f001:**
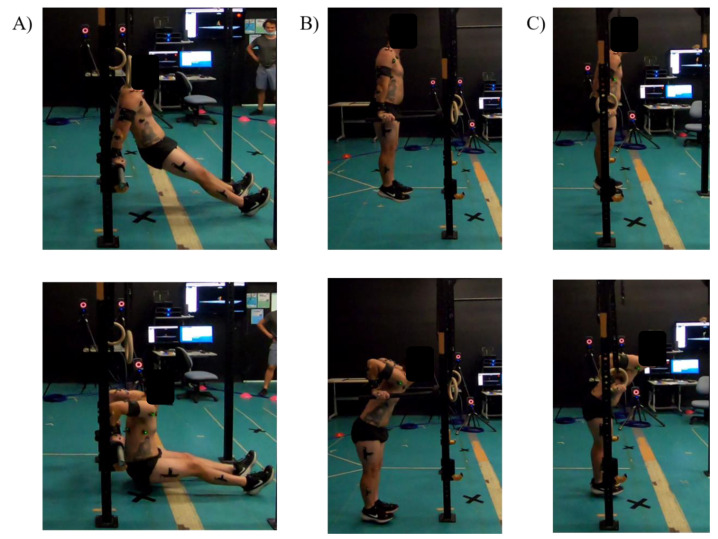
Typical starting position (**top** row) and bottom position (**bottom** row) for the (**A**) bench dip, (**B**) bar dip, and (**C**) ring dip.

**Figure 2 ijerph-19-13211-f002:**
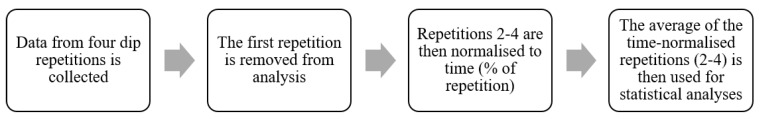
A flow chart representing the normalization procedures for each dip repetition and variation.

**Figure 3 ijerph-19-13211-f003:**
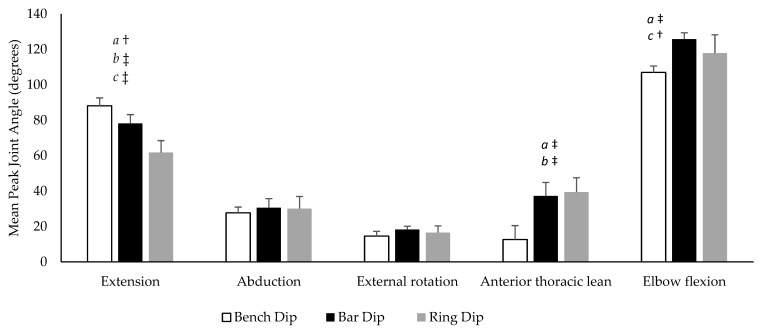
Mean peak joint angles during the bench, bar, and ring dips. a = significant difference between the bench dip and bar dip, b = significant difference between the bench dip and ring dip, c = significant difference between the bar dip and ring dip. † indicates large effect size (*d* > 0.8), ‡ indicates very large effect size (*d* > 1.2).

**Figure 4 ijerph-19-13211-f004:**
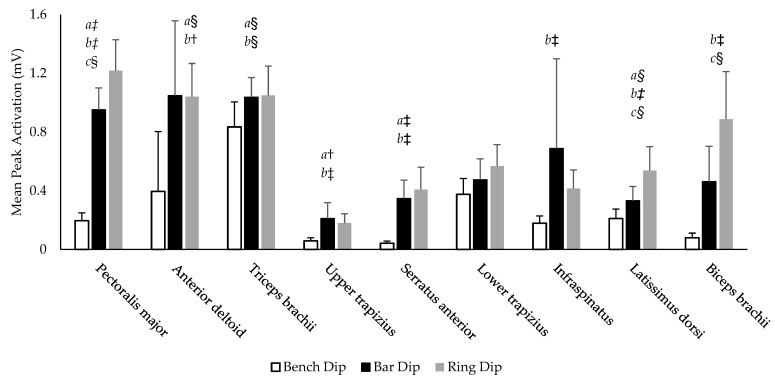
Mean peak sEMG amplitudes during the bench, bar, and ring dips. a = significant difference between the bench dip and bar dip, b = significant difference between the bench dip and ring dip, c = significant difference between the bar dip and ring dip. § indicates medium effect size (*d* > 0.5), † indicates large effect size (*d* > 0.8), ‡ indicates very large effect size (*d* > 1.2).

**Figure 5 ijerph-19-13211-f005:**
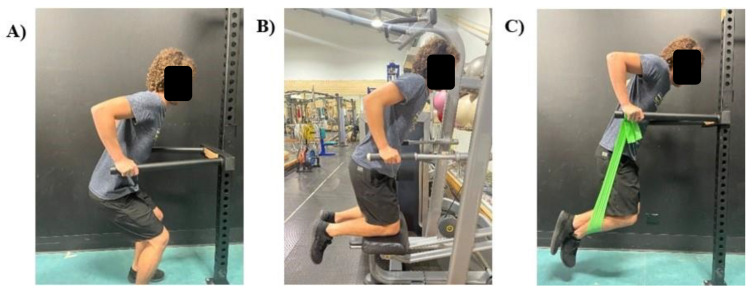
Examples of bar dip alternatives, that help exercisers moderate load at the point of maximal shoulder extension. (**A**) standing bar dips, (**B**) assisted bar dips, and (**C**) banded bar dips.

**Figure 6 ijerph-19-13211-f006:**
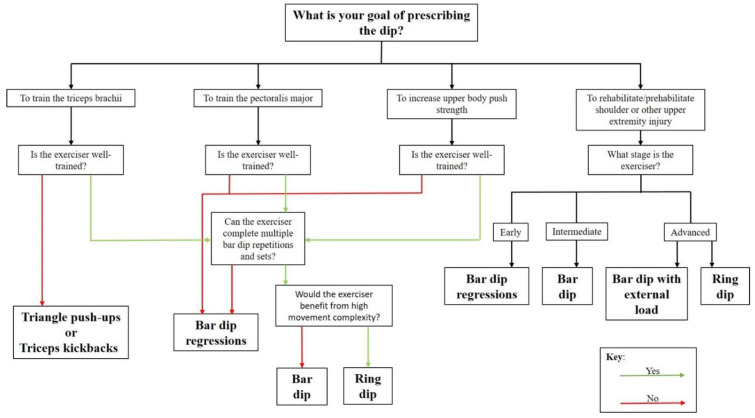
A flow chart identifying which dip variations should be prescribed for four common reasons.

## Data Availability

All data that support these findings are available from the corresponding author upon reasonable request.

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
