# Peer review of "Bench, Bar, and Ring Dips: Do Kinematics and Muscle Activity Differ?"

_ijerph, 2022, doi:10.3390/ijerph192013211_

Round 1

Reviewer 1 Report

The problem examined by the authors is interesting, especially in the case of rehabilitation procedures, the characteristics of the applied forces are important. However, the surveys were carried out correctly, and the description of the circumstances needs to be supplemented. The kinematic studies, also indicated in the title, also assume knowledge of the dynamics of displacement in space. I consider this to be important because in addition to sampling at 200 Hz, the values of speed and acceleration would also be included in the evaluation of data normalized over time. After the authors refer to the individual pace of execution, the effect of movement speed and EMG activity was not discussed. Of particular interest is the problem in the case of activity measured in the eccentric phase. The 0.5% representation of normalized data relative to the time of movement would deserve further elaboration.  

Beszúrás

Author Response

  • “The surveys were carried out correctly, and the description of the circumstances need to be supplemented”
    • The authors agree that the correct approach was used and believe that sufficient detail has been provided
  • “…the values of speed and acceleration would also be included in the evaluation…”
    • The authors agree that the movement speed and acceleration may be interesting to investigate. We feel this data would be most impactful when used to help determine tissue loading and tissue loading rates. However, these data would require data inputs such as force sensors or intertrial sensors that were not used in the current experiment. As this is the first article to investigate different dip variations in such depth, we feel that muscle activity to determine between-variation intensity and timing differences, together with joint displacements and ranges of motion, provides substantial insight regarding key aspects of the neuromechanical profiles. The data presented in this study provides strong indications regarding potential tissue strains at and near end range under different conditions. This is important information for those who are already prescribing the dip, to help them better determine circumstances when such prescriptions are potentially unsafe or inadvisable. We have concluded a recommendation regarding future investigations and data pertaining to movement dynamics in lines 320-321.

Reviewer 2 Report

This manuscript assess the upper limb muscles that are activated when subjects were doing 3 different types of upper arm dips.  This is an interesting study and the data are relevant to athletes, but maybe not to everyone.  The authors might want to add to their discussion and talk about how the various dips may affect older/non-athletes with shoulder conditions

Line 54: define "PM"

In figure 3, for elbow flexion, is the bench dip different than the ring dip or is it different than the bar dip?  It looks like it'd different than the bar dip.

Figure 5 seems out of place.  Variations in exercise are only briefly mentioned in a sentence in the "Conclusions" .  The authors may want to expand their discussion of variations for dips and put this into the regular discussion.

Author Response

  • “add to their discussion and talk about how the various dips may affect older/non-athletes with shoulder conditions”
    • The authors afree with this suggestion (we note, also shared by reviewer 3). We have added comments regarding potential limitations of this current research for older individuals and other populations with hypo- or hyper- mobility conditions, to lines 321-325 in the discussion section.
  • “line 54: define “PM”
    • Pectoralis major (PM) was previous defined in line 31-32
  • “In figure 3, for elbow flexion, is the bench dip different than the ring dip or is it different than the bar dip? It looks like it’d be different than the bar dip”
    • The authors acknowledge that upon visual inspection, the graphed data does give that impression. However, we have double-checked the statistics and the pairwise comparisons and can confirm they are correct. Elbow flexion: bench-bar (p=.026), bench-ring (p=.129), and bar-ring (p=.031).
  • “[referring to Figure 5] The authors may want to expand their discussion of variations for the dips and put this into their regular discussion.”
    • The authors acknowledge this is a good suggestion and have added comments regarding bar dip variations as regressions, in lines 273-283.

Reviewer 3 Report

Congratulations on your work,

I would like to make some considerations but I consider the authors have done a a good job  

INTRODUCTION

- Search bibliographic reference for the first paragraph

- I think a broader introduction should be introduced. Strength training programs/protocols could be described that have included this type of exercise with its corresponding muscle activation. Or other exercises that have similar activations of the most involved muscles (for example, the military press). Include references if there are studies that use the work in this exercise.  

MATERIALS AND METHODS

- Include study design

- Were protocols carried out prior to the study to ensure the same execution of all participants? This is essential to ensure homogeneity in the results.  RESULTS - I find this section very interesting and I think it is the one that has the greatest implication for the study, only in the figure would I put the letters (a-b-c) with all lowercase or uppercase. On the other hand, a very large effect is indicated in Extension and elbow flexion. Why do you think it may be due? And above all, what implications can this have?

- In the case of muscle activation results, it is quite clear that the ring dip was generally the most active, which was quite normal. What do you think is due to a greater activation of the infraspinatus in the "bar dip" ( and with so much deviation between the participants)

- I think that if the activation of the transversus abdominis muscle had been verified, in the case of the rings it would have been much higher than the rest.  

DISCUSSION

- Nice job and practical applications. I would also indicate as limitations the small study sample, the fact that there were only men (there were no women in the study) and that the age of the participants could perhaps modify the results (perhaps, with more adult-elderly ages due to mobility restriction , the results may have been different)  

Good luck. 

Author Response

INTRODUCTION

  • “Search bibliographic reference for the first paragraph”
    • The authors agree and have included a reference to two articles that describe the dip technique (line 29).
  • “Strength training programs/protocols could be described that have included this type of exercise with its corresponding muscle activation”
    • The authors agree and have included three articles that have used EMG to investigate the push-up, three which have investigated EMG during the shoulder press, and a systematic review investigating EMG and the bench press exercise. (lines 50-53)

MATERIALS AND METHODS

  • “Include study design”
    • The authors have now included a statement regarding the study in lines 80-81
  • “Were protocols carried out prior to the study to ensure the same execution of all participants?”
    • Yes they were. Extensive pilot testing was conducted to test specific wording in the instructions provided to participants. As only experienced exercisers were recruited, no misinterpretation of the exercise was observed in either pilot or actual data collection. We deliberately included no exercise standards/instructions so as not to influence the movement patterns of the participant. A standardized warm-up and familiarisation session was also performed prior to every testing session as outlines in lines 120-121.

RESULTS

  • “In the figure would I put the letter (a-b-c) with all lowercase or uppercase.”
    • The authors agree and both figures now contain all lower cases letters (a-b-c)
  • “On the other hand, a very large effect is indicated in Extension and elbow flexion. Why do you think it may be due? And above all, what implications can this have?”
    • The authors hypothesize the large changes observed in extension and elbow flexion relate primarily to two factors: (i) the amount of anterior thoracic lean, and (ii) the overall movement stability. Exercisers may attempt to maintain similar end-point depth, but are required to utilize a greater ROM in shoulder extension and elbow flexion when they are unable to lean the torso. Secondly, the decreased ROM observed in the ring dip may be due to the participant self-limiting the ROM so as not to load end-range in an unstable environment. Point (i) is described in discussion lines 232-234, and (ii) is described (and modified) in lines 305.
  • “What do you think is due to a greater activation of the infraspinatus in the “bar dip” (and with so much deviation between participants)”
    • This may be due to a low surface EMG reliability of infraspinatus and the high potential for cross-talk between surrounding muscles.

  • “I think if the activation of the transversus abdominis muscle had been verified, in the case of the rings it would have been much higher than the rest”
    • The authors agree with this statement. Transverse abdominus is a crucial muscle for inner core stability and one might reasonably assume that it would be most activated in the least stable condition. However, this muscle cannot be assessed reliably using surface EMG and we confined our investigation to muscles that had previously been investigated and that we determined were primarily involved in controlling and stabilising the shoulder complex, which contained the anatomy of focus for this study.

DISCUSSION

  • “I would also indicate as limitations the small study sample, the fact that there were only men (there were no women in the study) and that the age of the participants could perhaps modify the results”
    • The authors agress and have included relevant statements in lines 320-324